# Modelling Quintessential Inflation in Palatini-Modified Gravity



Konstantinos Dimopoulos [1,*], Alexandros Karam [2], Samuel Sánchez López [1] and Eemeli Tomberg [2]

1   Consortium for Fundamental Physics, Physics Department, Lancaster University, Lancaster LA1 4YB, UK; s.sanchezlopez@lancaster.ac.uk
2   Laboratory of High Energy and Computational Physics, National Institute of Chemical Physics and Biophysics, Rävala pst. 10, 10143 Tallinn, Estonia; alexandros.karam@kbfi.ee (A.K.); eemeli.tomberg@kbfi.ee (E.T.)
*   Correspondence: konst.dimopoulos@lancaster.ac.uk

**Abstract:** We study a model of quintessential inflation constructed in $R^2$-modified gravity with a non-minimally coupled scalar field, in the Palatini formalism. Our non-minimal inflaton field is characterised by a simple exponential potential. We find that successful quintessential inflation can be achieved with no fine-tuning of the model parameters. Predictions of the characteristics of dark energy will be tested by observations in the near future, while contrasting with existing observations provides insights on the modified gravity background, such as the value of the non-minimal coupling and its running.

**Keywords:** cosmology; modified gravity; cosmic inflation; dark energy; quintessence; palatini gravity





## 1. Introduction

Observations suggest that the Universe has undergone at least two phases of accelerated expansion. The primordial phase is called cosmic inflation and it is responsible for arranging the fine-tuning needed for the subsequent hot big bang evolution of the Universe, as well as for generating cosmological perturbations, which are responsible for structure formation [1]. The late phase is taking place at present and it is attributed to the gradual dominance of the mysterious dark energy substance, which makes up almost 70% of the Universe's content today [2].

In the context of fundamental theory, cosmic inflation is typically realised according to the inflationary paradigm, which suggests that the expansion of the Universe is accelerating when the latter is dominated by the potential density of a scalar field, called the inflaton field. Similarly, dark energy can also be modelled as a suitable scalar field, called quintessence [3]. It is natural to attempt to unify the two phases and consider that accelerated expansion in the Universe is due to a single agent. The proposal is called quintessential inflation [4].

Apart from being economic, quintessential inflation addresses holistically accelerated expansion in the early and late Universe in a single theoretical framework. A successful quintessential inflation model has to satisfy the observations of both inflation and dark energy. As such, constructing a quintessential inflation model is highly constrained and very difficult to achieve, but not impossible (e.g., see Refs. [5–9] for recent reviews).

From the very beginning, modelling cosmic inflation was attempted in modified gravity as well as particle physics. Indeed, the very first inflation model was Starobinsky's $R^2$ inflation [10]. It is harder to use modified gravity for dark energy, however, because deviation from Einstein's general relativity should not violate stringent constraints set by a plethora of experiments (solar system, Eötvös, etc.). This is why, in attempting to construct a quintessential inflation model, we assume a blended approach, where modified gravity is mainly employed for inflation, while particle theory (which is behind our scalar potential) accounts for dark energy.

In our model, we consider the Palatini formulation of gravity [11,12]. In the Palatini formulation, the connection and the metric are independent variables. In general relativity

the traditional metric formalism and the Palatini formalism are equivalent. However, this is not so when matter is non-minimally coupled to gravity or when the action is no longer linear in $R$, the curvature scalar.

Metric $R^2$ gravity introduces a new degree of freedom (dof), which can be expressed as a scalar field (scalaron) in the Einstein frame [10]. In contrast, Palatini $R^2$ gravity has no extra propagating dof that can play the role of the inflaton field. Therefore, an additional scalar field must be introduced.

In Palatini $R^2$ inflation, one can lower the tensor-to-scalar ratio in any scalar field inflation model [13,14]. Moreover, Palatini-modified gravity evades the stringent constraints on the propagation speed of primordial gravitational waves. Finally, Palatini gravity does not suffer that much from solar system and other related bounds on modified gravity, which means it is ideal for modelling quintessential inflation [15].

In quintessential inflation, the thermal bath of the hot big bang is not generated by the decay of the inflaton field, because the latter must survive until the present to become quintessence. An alternative mechanism for reheating the Universe must be employed. In this work, we do not consider a specific mechanism for reheating the Universe. However, there is a plethora of such mechanisms [16–21] (see also Refs. [22,23]), and we assume the operation of one of them.

We use natural units, where $c = \hbar = 1$ and $8\pi G = m_P^{-2}$ with $m_P = 2.43 \times 10^{18}\,\text{GeV}$ being the reduced Plank mass.

## 2. The Model

We consider the action in the Palatini formalism

$$S = \int d^4x \sqrt{-g}\left[\frac{1}{2}m_P^2 F(\varphi, R) - \frac{1}{2}g^{\mu\nu}\partial_\mu\varphi\partial_\nu\varphi - V(\varphi)\right] + S_m[g_{\mu\nu}, \psi], \tag{1}$$

where $\psi$ collectively represents the matter fields other than the inflaton. The function $F(\varphi, R)$ takes the following form[1]

$$F(\varphi, R) = \left(1 + \xi\frac{\varphi^2}{m_P^2}\right)R + \frac{\alpha}{2m_P^2}R^2, \tag{2}$$

with $R$ being the Ricci scalar, which is a function of the connection only

$$R = g^{\mu\nu}R_{\mu\nu}(\Gamma). \tag{3}$$

Note that both terms in Equation (2) are well motivated in the literature since they can naturally arise when one considers quantum corrections (e.g., see Ref. [27]). The above action is dynamically equivalent to

$$\begin{aligned}S = &\int d^4x \sqrt{-g}\left[\frac{1}{2}m_P^2\left(1 + \xi\frac{\varphi^2}{m_P^2} + \frac{\alpha}{m_P^2}\chi\right)R - \frac{1}{4}\alpha\chi^2 - \frac{1}{2}g^{\mu\nu}\partial_\mu\varphi\partial_\nu\varphi - V(\varphi)\right]\\ &+ S_m[g_{\mu\nu}, \psi],\end{aligned} \tag{4}$$

where $\chi$ is an auxiliary scalar field, which will be dispensed below.

To assist our intuition, we switch to the Einstein frame by a suitable conformal transformation

$$g_{\mu\nu} \to \bar{g}_{\mu\nu} = \left(1 + \xi\frac{\varphi^2}{m_P^2} + \frac{\alpha}{m_P^2}\chi\right)g_{\mu\nu}. \tag{5}$$

Now we eliminate the auxiliary field by obtaining its equation of motion

$$\frac{\delta S}{\delta \chi} = 0 \quad \Leftrightarrow \quad \chi = \frac{4m_P^2 V + (m_P^2 + \xi\varphi^2)(\bar{\partial}\varphi)^2}{(m_P^2 + \xi\varphi^2)m_P^2 - \alpha(\bar{\partial}\varphi)^2}, \tag{6}$$

where $(\bar{\partial}\varphi)^2 \equiv \bar{g}^{\mu\nu}\bar{\partial}_\mu\varphi\bar{\partial}_\nu\varphi$. Substituting $\chi$ back into the action yields

$$
S_E = \int \mathrm{d}^4x \sqrt{-\bar{g}} \left[ \frac{1}{2}m_P^2\bar{R} - \frac{1}{2}(\partial\varphi)^2 \frac{1 + \frac{\xi\varphi^2}{m_P^2}}{\left(1 + \frac{\xi\varphi^2}{m_P^2}\right)^2 + \frac{4\alpha V(\varphi)}{m_P^4}} \right.
$$
$$
\left. - \frac{V(\varphi)}{\left(1 + \frac{\xi\varphi^2}{m_P^2}\right)^2 + \frac{4\alpha V(\varphi)}{m_P^4}} + \frac{1}{4}\frac{\alpha}{m_P^4} \frac{(\partial\varphi)^4}{\left(1 + \frac{\xi\varphi^2}{m_P^2}\right)^2 + \frac{4\alpha V(\varphi)}{m_P^4}} \right] + S_m[\Omega^{-2}\bar{g}_{\mu\nu}, \psi]. \tag{7}
$$

Note that in the Palatini formalism the auxiliary field is not dynamical, which allowed us to use its equation of motion to eliminate it. Thus, the resulting action only contains one scalar field, albeit with non-canonical kinetic terms and a modified potential. This is in contrast to the metric version of the theory, where the auxiliary field has its own kinetic term and the resulting action is two-field.

The canonical field $\phi$ is obtained via the redefinition

$$
\frac{\mathrm{d}\phi}{\mathrm{d}\varphi} = \sqrt{\frac{1 + \frac{\xi\varphi^2}{m_P^2}}{\left(1 + \frac{\xi\varphi^2}{m_P^2}\right)^2 + \frac{4\alpha V(\varphi)}{m_P^4}}}. \tag{8}
$$

One can use $\varphi = \varphi(\phi)$ to obtain the potential in the Einstein frame

$$
\bar{V}(\phi) = \frac{V(\phi)}{\left(1 + \frac{\xi\varphi^2(\phi)}{m_P^2}\right)^2 + \frac{4\alpha V(\phi)}{m_P^4}}. \tag{9}
$$

The above suggests that, for very large values of $V(\phi)$, the term in brackets in the denominator becomes negligible and the overall potential in the Einstein frame approximates a constant given by $\bar{V} \to m_P^4/4\alpha$. This is the inflationary plateau, attained regardless of the specific form of $V(\varphi)$ as long as the latter becomes very large in some limit.

Regarding the quintessential tail, the second flat region of the scalar potential, which is responsible for dark energy at present, we note that near the present time, $R$ is tiny, so the $\alpha R^2$ term in the Lagrangian is negligible. This is equivalent to setting $\alpha = 0$. In this case, Equation (8) reduces to

$$
\frac{\mathrm{d}\phi}{\mathrm{d}\varphi} = \frac{1}{\sqrt{1 + \frac{\xi\varphi^2}{m_P^2}}} \tag{10}
$$

which results in

$$
\varphi = \frac{m_P}{\sqrt{\xi}} \sinh\left(\frac{\sqrt{\xi}\,\phi}{m_P}\right). \tag{11}
$$

We consider that the runaway potential of the inflaton/quintessence scalar field $\varphi$ is

$$
V(\varphi) = M^4 e^{-\kappa\varphi/m_P}, \tag{12}
$$

where the dimensionless constant $\kappa$ is the strength of the exponential and $M$ is an energy scale. An exponential potential is well motivated in particle physics. Using the above and

Equations (9) and (10) with $\alpha = 0$, we find that the scalar potential of the quintessential tail in the Einstein frame is

$$\bar{V}(\phi) = M^4 \frac{\exp\left[-\frac{\kappa}{\sqrt{\xi}}\sinh\left(\frac{\sqrt{\xi}\,\phi}{m_P}\right)\right]}{\cosh^4\left(\frac{\sqrt{\xi}\,\phi}{m_P}\right)}. \tag{13}$$

Note that, in the limit $\sqrt{\xi}\,\phi \ll m_P$, the above reduces to $\bar{V} = M^4 \exp(-\kappa\,\phi/m_P)$. In this limit, Equation (8) suggests $\phi \approx \varphi$, i.e., $\varphi$ is approximately canonical. Thus, in this limit we end up with the usual exponential quintessential tail, which leads to accelerated expansion only if $\kappa < \sqrt{2}$. When $\kappa$ is larger, the exponential potential leads to the scaling solution, which cannot result in accelerated expansion. However, as $\phi$ grows, the Einstein frame potential becomes steeper than an exponential and accelerated expansion ceases, even if $\kappa$ is small enough.

If $\kappa$ is small enough to lead to accelerated expansion when $\phi$ is small, then inflation would not be able to end even after the field exits the inflationary plateau. This is why we consider the effect of the $\alpha R^2$ term, which is negligible at late times, but important at early times and high energies. Thereby, we can facilitate a graceful exit from inflation and still achieve accelerated expansion at present. However, we find that the value of the non-minimal coupling $\xi$ is not the same for successful inflation and quintessence. Therefore, we consider a mild running of $\xi$ as follows

$$\xi(\varphi) = \xi_*\left[1 + \beta\ln\left(\frac{\varphi^2}{\mu^2}\right)\right], \tag{14}$$

where $\mu$ is a suitable mass scale, and $\xi_*$ and $\beta$ are constants, to be determined by the observations. The above is suggested by renormalisation considerations. The scalar field only slowly varies (rolls) when the cosmological scales exit the horizon during inflation and also when quintessence thaws while dominating the Universe at present. This means that, in both cases, $\xi \simeq$ constant. However, because $\varphi$ changes dramatically between inflation and quintessence, the non-minimal coupling is not going to be the same in both cases.

The scalar potential in the Einstein frame is depicted in Figure 1.

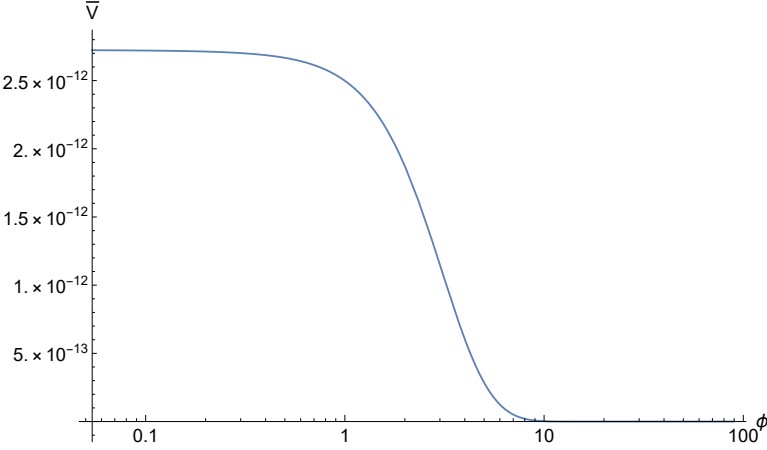

**Figure 1.** The scalar potential $\bar{V}$ in Planck units in the Einstein frame, featuring the inflationary plateau and the quintessential tail.

### 3. Equation of Motion

Varying the action with respect to $\varphi$, we have

$$\ddot{\varphi} + 3H\dot{\varphi} + V'(\varphi) - \left[\xi(\varphi) + \frac{1}{2}\xi'(\varphi)\varphi\right]\varphi R = 0, \tag{15}$$

which, using Equation (14), immediately reads

$$\ddot{\varphi} + 3H\dot{\varphi} + V'(\varphi) = \xi_* \left[ 1 + \beta \left( 1 + \ln \frac{\varphi^2}{\mu^2} \right) \right] \varphi R, \tag{16}$$

where the prime denotes a derivative with respect to the argument ($\varphi$ in this case) and the dot denotes a derivative with respect to time in the Jordan frame.

To investigate $R$, we need to consider the energy-momentum tensor. We have

$$T_{\mu\nu} = -\frac{2}{\sqrt{-g}} \frac{\delta S}{\delta g^{\mu\nu}} = \left( F_R R_{(\mu\nu)} - \frac{1}{2} g_{\mu\nu} F \right) m_P^2, \tag{17}$$

where $F_R(\varphi, R) \equiv \partial_R F(\varphi, R)$ with $F(\varphi, R)$ given in Equation (2). The trace of the above is $T = (F_R R - 2F) m_P^2$. Thus, the (Palatini) Ricci scalar is algebraically related to the matter sources as

$$R = \frac{1}{m_P^2 + \xi \varphi^2} \left[ \rho(1 - 3w) - \dot{\varphi}^2 + 4V(\varphi) \right], \tag{18}$$

where we used that the trace of the energy-momentum tensor is [28]

$$T = \dot{\varphi}^2 - 4V(\varphi) - \rho(1 - 3w), \tag{19}$$

with $\rho$ the energy density and $w$ the barotropic parameter of the background matter, dominant or not. Note that, when the background matter is dominant, then $T = (3w - 1)\rho$, which is zero during radiation domination, since then $w = \frac{1}{3}$. The same is true of $R$ itself. As a result, during radiation domination the equation of motion of $\varphi$ reduces to the usual Klein–Gordon of a minimally coupled scalar field. It is also interesting that both $R$ and $T$ above are independent from the value of $\alpha$.[2]

In the Einstein frame, there is a new coupling between the matter action and the inflaton field. Indeed, its equation of motion now reads [28]

$$\frac{\delta S_E}{\delta \phi} = \sqrt{-\bar{g}}(\ddot{\phi} + 3H\dot{\phi}) + \frac{\mathrm{d}\varphi}{\mathrm{d}\phi} \left( \sqrt{-\bar{g}} \bar{V}'(\varphi) + \frac{\delta S_m}{\delta \varphi} \right) = 0, \tag{20}$$

where $\mathrm{d}\varphi/\mathrm{d}\phi$ is given by Equation (8). The functional derivative of the matter action is [28]

$$\frac{\delta S_m}{\delta \varphi} = \frac{\sqrt{-\bar{g}}\, \xi \varphi}{m_P^2 + \xi \varphi^2 - \alpha(\bar{\partial}\varphi)^2/m_P^2} \bar{\rho}(1 - 3\bar{w}), \tag{21}$$

where $\bar{w}$ is the barotropic parameter of the background matter (assumed to be a barotropic ideal fluid), which is the same in both the Einstein and the Jordan frames $\bar{w} = w$ [28]. From the above we see that, when the background matter is radiation (dominant or not), for which $w = \frac{1}{3}$, then the coupling of the inflaton to matter disappears. Thus, this coupling is only effective after matter–radiation equality, when the Universe is matter-dominated. As we have discussed, at late times the contribution of the $\alpha R^2$ term in the Lagrangian density is negligible. This is equivalent to setting $\alpha \to 0$ in the above.

Regarding the derivative of the potential in Equation (20) (but neglecting the running of $\xi$ as subleading), we have

$$\frac{\mathrm{d}\bar{V}(\varphi)}{\mathrm{d}\varphi} = \frac{V'(\varphi)}{\left( 1 + \frac{\xi \varphi^2}{m_P^2} \right)^2 + \frac{4\alpha V}{m_P^4}} - \frac{V(\varphi) \left[ \frac{4\alpha}{m_P^4} V'(\varphi) + \frac{4\xi \varphi}{m_P^2} \left( 1 + \frac{\xi \varphi^2}{m_P^2} \right) \right]}{\left[ \left( 1 + \frac{\xi \varphi^2}{m_P^2} \right)^2 + \frac{4\alpha V}{m_P 2^4} \right]^2}, \tag{22}$$

where we considered Equation (9).

## 4. Inflation

Inflation is expected to occur when we are on the inflationary plateau (in the Einstein frame) with a large value of $V$, i.e., $\varphi \ll 0$. In this limit, we can consider slow-roll inflation in the Einstein frame, which is determined by the slow-roll parameters

$$\epsilon \equiv \frac{1}{2}\left(\frac{\mathrm{d}\bar{V}}{\mathrm{d}\phi}\frac{m_P}{\bar{V}}\right)^2 \quad \text{and} \quad \eta \equiv \frac{\mathrm{d}^2\bar{V}}{\mathrm{d}\phi^2}\frac{m_P^2}{\bar{V}}. \tag{23}$$

To have slow-roll, $\epsilon < 1$ and $|\eta| < 1$. Ref. [13] suggests that the above are given by

$$\epsilon = \frac{\tilde{\epsilon}}{1+4\alpha\tilde{V}} \quad \text{with} \quad \tilde{\epsilon} = \frac{1}{2}\frac{\left[\kappa\left(1+\frac{\xi\varphi^2}{m_P^2}\right)+4\xi\frac{\varphi}{m_P}\right]^2}{1+\frac{\xi\varphi^2}{m_P^2}}, \tag{24}$$

$$\eta = \tilde{\eta} - 3\frac{4\alpha\tilde{V}}{1+4\alpha\tilde{V}} \quad \text{with} \quad \tilde{\eta} = \frac{\left(7\kappa\xi\frac{\varphi}{m_P}+\kappa^2\right)\left(1+\frac{\xi\varphi^2}{m_P^2}\right)-4\xi+16\xi^2\frac{\varphi^2}{m_P^2}}{1+\frac{\xi\varphi^2}{m_P^2}}, \tag{25}$$

where the tilded quantities correspond to $\alpha = 0$ (and we have taken the limit of constant $\xi$). In the above

$$\tilde{V} \equiv \frac{M^4 e^{-\kappa\varphi/m_P}}{(1+\frac{\xi\varphi^2}{m_P^2})^2} \tag{26}$$

which is readily obtained by Equations (9) and (12) when $\alpha \to 0$. To contrast with observations, we obtain the standard inflationary observables

$$n_s = 1 - 6\epsilon + 2\eta = 1 - 6\tilde{\epsilon} + 2\tilde{\eta}, \tag{27}$$

$$r = 16\epsilon \quad \text{and} \quad 24\pi^2 m_P^4 A_s = \frac{\bar{V}}{\epsilon} = \frac{\tilde{V}}{\tilde{\epsilon}}, \tag{28}$$

where $A_s$ is the scalar power spectrum, $n_s$ is the spectral index and $r$ is the tensor-to-scalar ratio at the CMB pivot scale $k_* = 0.05\,\mathrm{Mpc}^{-1}$. In the above, we used that $\bar{V}/\epsilon = \tilde{V}/\tilde{\epsilon}$ as shown in Ref. [13]. Technically, $\beta \neq 0$, so Equations (27) and (28) are not exact, but we expect the modification to be minor because the non-minimal coupling depends only logarithmicaly on the slowly rolling inflaton (see Equation (14)).

The observations suggest [29,30]

$$\ln\left(10^{10}A_s\right) = 3.044 \pm 0.014, \quad n_s = 0.9649 \pm 0.0042, \quad r < 0.036. \tag{29}$$

From this and Equation (28), it is straightforward to find

$$2.1 \times 10^{-9} = A_s = \frac{M^4 e^{-\kappa\varphi_*/m_P}}{12\pi^2 m_P^4\left(1+\frac{\xi\varphi_*^2}{m_P^2}\right)\left[\kappa\left(1+\frac{\xi\varphi_*^2}{m_P^2}\right)+4\xi\varphi_*\right]^2}, \tag{30}$$

where the subscript '$*$' denotes the exit of the pivot scale during inflation and we employed Equations (24) and (26). From Equations (24), (25) and (27), the spectral index is

$$n_s - 1 = -\kappa^2\left(1+\frac{\xi\varphi_*^2}{m_P^2}\right) - 10\,\xi\kappa\frac{\varphi_*}{m_P} - 8\xi\frac{1+2\left(\frac{\xi\varphi_*^2}{m_P^2}\right)}{1+\frac{\xi\varphi_*^2}{m_P^2}}. \tag{31}$$

Finally, for the number of e-folds we have

$$
\begin{aligned}
N &= -\frac{1}{m_P} \int \frac{\mathrm{d}\phi}{\sqrt{2\epsilon}} = -\frac{1}{m_P} \int \frac{\mathrm{d}\varphi}{\kappa \left(1 + \frac{\xi\varphi^2}{m_P^2}\right) + 4\frac{\xi\varphi}{m_P}} \\
&= -\frac{1}{\sqrt{\xi\kappa^2 - 4\xi^2}} \arctan\left[\frac{\sqrt{\xi}\left(2 + \kappa\frac{\varphi}{m_P}\right)}{\sqrt{\kappa^2 - 4\xi}}\right].
\end{aligned}
\tag{32}
$$

## 5. Kination

At some point the inflationary plateau ends and the potential becomes steep and curved. Inflation ends and the inflaton field falls over a potential cliff. As a result, the kinetic energy density of the field dominates the Universe. The Palatini Ricci tensor in Equation (18) becomes $R = -\dot\varphi^2/(m_P^2 + \xi\varphi^2)$. In principle, the quartic kinetic term in Equation (7) might affect the dynamics of kination, but we find otherwise (see Figure 2). In addition, as we have seen, the coupling of the inflaton to the background matter disappears if the background is radiation. Thus, kination proceeds as usual, with $\rho \propto a^{-6}$ and $a \propto t^{1/3}$ [31].

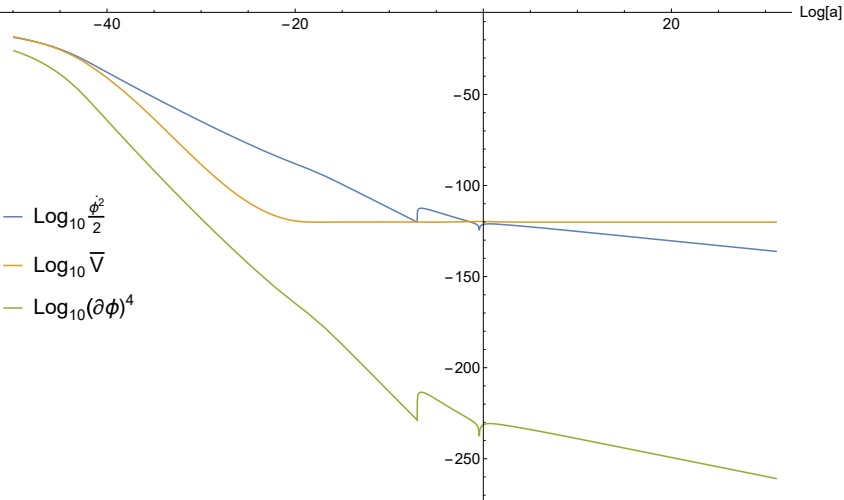

**Figure 2.** Log-log plot in Planck units of the contributions to the inflaton energy density as a function of the scale factor normalised to unity at present, starting at the end of inflation. The upper curve (blue) corresponds to the canonical kinetic energy density of the scalar field, which dominates the potential until the time of equal-matter-radiation densities (equality). At this time the field briefly freezes only to unfreeze in matter domination (because of the interaction with matter in Equation (21)) and again becomes dominated by its kinetic energy density until the present time when the potential energy density, depicted by the middle curve (orange), takes over. The quartic kinetic term in Equation (7), depicted by the lower curve (green), always remains negligible.

We assume that subdominant radiation is generated at the end of inflation (denoted by the super/subscript 'end'), with density parameter $\Omega_r^{\mathrm{end}} = (\rho_r/\rho_{\mathrm{tot}})_{\mathrm{end}}$, which is also called the reheating efficiency. The density of the background radiation scales as $\rho_r \propto a^{-4}$. This means that there is a moment when $\rho_r$ becomes dominant over the rolling scalar field and the Universe becomes radiation-dominated. This is the moment of reheating. After reheating, the field continues to roll kinetically dominated for a while until its potential density becomes important. If the slope of the latter is small enough, the field freezes.

Things change after matter–radiation equality, when the interaction of the field (which is now quintessence) with matter affects its dynamics. We find that quintessence unfreezes and rolls further, until it dominates the Universe again. The evolution of the energy density of the scalar field and of the background density is shown in Figure 3.

The early, stiff kination era increases the number of e-folds between the end of inflation and the horizon exit of the CMB scale from the standard 50–60 to 60–75. We have taken the full expansion history into account when fixing the CMB scale in our results.

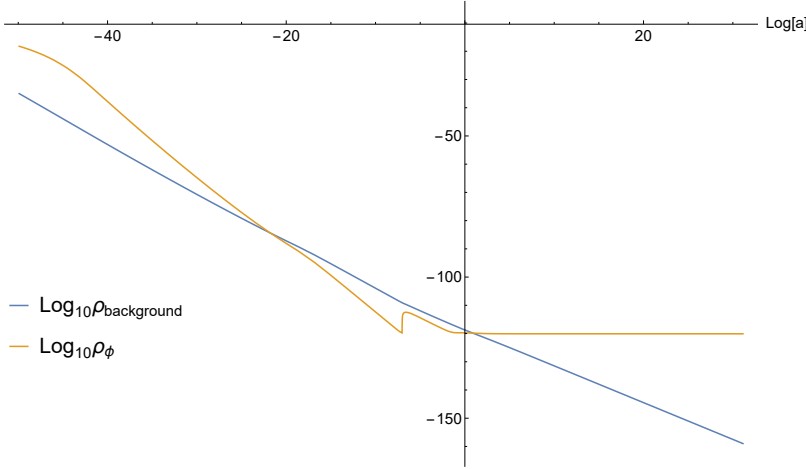

**Figure 3.** Log-log plot of the energy density of the scalar field in Planck units (orange) and that of the background (blue) after the end of inflation. Originally, the scalar field kinetic density dominates (kination) until the moment of reheating when the background density takes over. The scalar field momentarily freezes at equality, but then unfreezes in the matter era, due to the background backreaction. Eventually, it comes to dominate at present.

## 6. Quintessence

Soon after matter–radiation equality, quintessence refreezes at some value $\phi_0$ (or $\varphi_0$ in terms of the non-canonical field). Then there are certain requirements it must satisfy if it is to be the observed dark energy, akin to the CMB observational constraints for inflation. The first such constraint is coincidence. This means that the density parameter of the frozen quintessence at present must be [32]

$$\Omega_\phi = \Omega_{\mathrm{DE}} = 0.6847 \pm 0.0073 \,. \tag{33}$$

In general, the barotropic parameter of quintessence is variable. By Taylor expanding it near the present, this varying barotropic parameter can be approximated as (CPL parametrisation [33,34])

$$w_{\mathrm{DE}} = w_{\mathrm{DE}}^0 + w_{\mathrm{a}}\left(1 - \frac{a}{a_0}\right), \tag{34}$$

where $w_{\mathrm{DE}}^0$ is the value of $w_{\mathrm{DE}}$ at present and

$$w_{\mathrm{a}} = -\left.\frac{\mathrm{d}w_{\mathrm{DE}}}{\mathrm{d}a}\right|_{a_0} = -\left.\frac{\mathrm{d}w_{\mathrm{DE}}}{\mathrm{d}t}\frac{1}{\dot{a}}\right|_{t_0}, \tag{35}$$

where '0' denotes the present time. Observations require [32]

$$-1 \leq w_{\mathrm{DE}}^0 < -0.95 \quad \text{and} \quad w_{\mathrm{a}} \in [-0.55, 0.03] \,. \tag{36}$$

Demanding that quintessence is successful dark energy implies that $w_\phi = w_{\mathrm{DE}}$, which must satisfy the above constraints.

Starting with the coincidence requirement, the quintessence density at present is

$$\rho_\phi^0 = 3H_0^2 m_P^2 \Omega_\phi = 3H_0^2 m_P^2 \Omega_{\mathrm{DE}} \approx 8 \times 10^{-121} m_P^4 \,, \tag{37}$$

where we approximated $H_0 \approx 67.8$ km/s/Mpc and we used Equation (33). Equation (26) suggests

$$\rho_\phi^0 \simeq \bar{V}(\varphi_0) = \frac{M^4 \, e^{-\kappa\varphi_0/m_P}}{\left(1 + \frac{\xi\varphi_0^2}{m_P^2}\right)^2} = M^4 \, \frac{e^{-\frac{\kappa}{\sqrt{\xi}}\sinh(\sqrt{\xi}\phi_0/m_P)}}{\cosh^4(\sqrt{\xi}\phi_0/m_P)}, \tag{38}$$

where we considered Equation (13) because $\alpha$ is negligible at late times. In the above, $\xi$ is not the same as in inflation, but it is given by Equation (14) as $\xi = \xi_*[1 + \beta\ln(\varphi_0^2/\mu^2)]$. We have also taken into account that the field is thawing so that its kinetic energy density is subdominant to its potential energy density and so $\rho_\phi \simeq V$. Because $\xi$ is logarithmically dependent on $\varphi$ and the latter varies mildly as quintessence thaws, we expect $\xi \simeq$ constant.

The value of $M$ is determined by the normalisation of the scalar perturbations during inflation:

$$A_s = \frac{2V(\varphi_*)}{3\pi^2 m_P^4 r}. \tag{39}$$

We further consider $|\xi| = |\xi(\varphi_0)| \ll 1$. In this limit, Equations (38) and (39) suggest

$$\frac{\kappa\varphi_0}{m_P} = -\left[\ln\left(\frac{\rho_\phi^0}{m_P^4}\right) - \ln\left(\frac{3\pi^2}{2}A_s r\right)\right] + \frac{\kappa\varphi_*}{m_P} \approx 252 + \ln\left(\frac{r}{10^{-3}}\right) + \frac{\kappa\varphi_*}{m_P}. \tag{40}$$

Because we find that $\kappa\varphi_* \sim -(\text{a few}) \times m_P$, we expect $\kappa\varphi_0/m_P \simeq 250$ or so. Using Equation (11), we find

$$\frac{\sqrt{\xi}\,\phi_0}{m_P} \simeq \sinh^{-1}\left\{\frac{\sqrt{\xi}}{\kappa}\left[252 + \ln\left(\frac{r}{10^{-3}}\right)\right] + \sinh\left(\frac{\sqrt{\xi}\,\phi_*}{m_P}\right)\right\}. \tag{41}$$

## 7. Results

The parameter space for successful inflation is shown in Figure 4. From this figure it is evident that inflation requires (see Table 1, for exact values used in the figures)

$$\kappa \approx 0.3 \quad \text{and} \quad \xi_* \approx 0.01 \tag{42}$$

where, without loss of generality, we choose that $\mu^2 \approx \varphi_*^2$ in Equation (14) such that, when the cosmological scales leave the horizon, we have $\xi \approx \xi_*$. We find that $\mu \simeq -6\,m_P$ (for $\mu = -6.00\,m_P$ we find $\varphi_* = -5.91\,m_P$). For the $\alpha$ parameter, we obtain a lower bound $\alpha \gtrsim 10^7$. We choose $\alpha \simeq 10^{11}$. The energy scale at the end of inflation is found to be $\bar{V}_{\text{end}}^{1/4} \simeq 3 \times 10^{-5}\,m_P \sim 10^{14}\,\text{GeV}$, which is somewhat smaller than the estimate of the inflationary plateau $\bar{V}^{1/4} = (4\alpha)^{-1/4}m_P \sim 10^{-3}\,m_P$. Similarly, the density scale in the scalar potential is $M^4 \simeq 2 \times 10^{-9}m_P^4$, which implies $M = 2 \times 10^{16}\,\text{GeV}$, i.e., the scale of grand unification.

For successful quintessence we consider the reheating efficiency $\Omega_r^{\text{end}} \sim 10^{-15}$. This value belongs comfortably in the allowed range,

$$10^{-2}\left(\frac{\bar{H}_{\text{end}}}{m_P}\right)^2 \sim 10^{-20} < \Omega_r^{\text{end}} < 1, \tag{43}$$

where the upper bound corresponds to prompt reheating, while the lower bound corresponds to gravitational reheating, for which $\rho_r^{\text{end}} \sim 10^{-2}\bar{H}_{\text{end}}^4$ [35,36]. In Equation (43) we used $\bar{H}_{\text{end}} \simeq \sqrt{(\bar{V}_{\text{end}}/3)}/m_P \sim 10^{-9}\,m_P$. There are many possibilities for reheating the Universe without the decay of the inflaton field, which are typically considered in non-oscillatory inflationary models. Examples are instant preheating [16], curvaton reheating [17,18] and Ricci reheating [19–21].

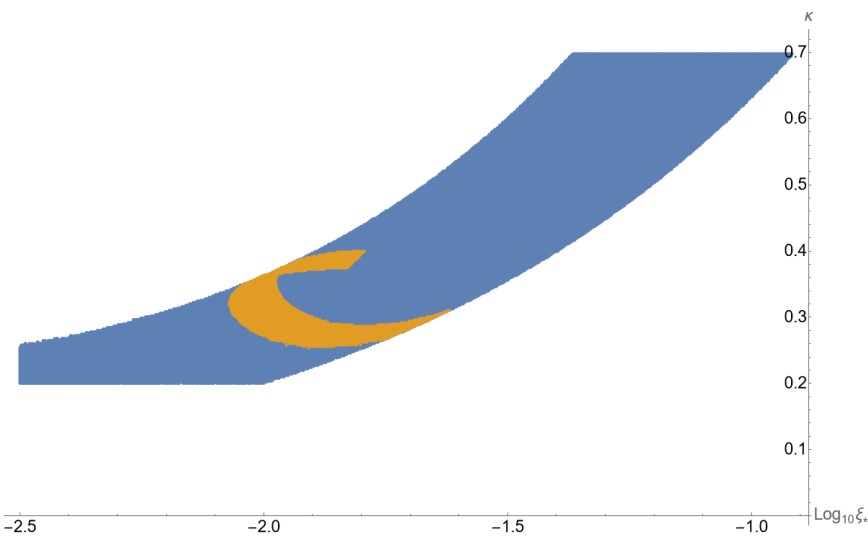

**Figure 4.** The parameter space $\kappa(\xi_*)$ for successful inflation. The blue (dark) band depicts the region which reproduces the observed values of the spectral index and the amplitude of the cosmological perturbations (the central values of $n_s$ and $A_s$ in Equation (29)). The allowed region is depicted in orange (light band), which satisfies the bound on the tensor-to-scalar ratio $r$ in Equation (29) and corresponds to the range of reheating efficiency in Equation (43), which in turn implies the number of efolds of remaining inflation when the cosmological scales leave the horizon ranges as $N_* = 60\text{–}75$. (We have taken $\alpha M^4 = 143.08$.).

**Table 1.** Exact values of model parameters assumed in all the figures.

| $\alpha = 9.16 \times 10^{10}$ | $\kappa = 0.2956$ | $M^4 = 2.11 \times 10^{-9}\, m_P^4$ |
|---|---|---|
| $\xi_* = 0.0093282$ | $\beta = -0.10075$ | $\mu = -6\, m_P$ |

Let us estimate the reheating temperature. Assuming proper kination begins right away after the end of inflation we find the following. During kination, the total energy density of the Universe decreases as $\rho_{\text{tot}} \simeq \rho_\phi \propto a^{-6}$, while for radiation we have $\rho_r \propto a^{-4}$, which means that $\rho_r/\rho_{\text{tot}} \propto a^2$. Therefore,

$$\Omega_r^{\text{end}} = \left.\frac{\rho_r}{\rho_{\text{tot}}}\right|_{\text{end}} = \left(\frac{a_{\text{end}}}{a_{\text{reh}}}\right)^2 \left.\frac{\rho_r}{\rho_{\text{tot}}}\right|_{\text{reh}} \simeq \left(\frac{a_{\text{end}}}{a_{\text{reh}}}\right)^2 , \tag{44}$$

where 'reh' denotes reheating, which is the moment that radiation takes over and we have $\rho_r \simeq \rho_{\text{tot}}$. The density of the Universe at reheating is straightforward to find, by considering that $\rho_{\text{tot}} \propto a^{-6}$. Indeed, we obtain

$$\rho_{\text{reh}} = \left(\frac{a_{\text{end}}}{a_{\text{reh}}}\right)^6 \rho_{\text{end}} \simeq (\Omega_r^{\text{end}})^3 \, \bar{V}_{\text{end}} , \tag{45}$$

where we used Equation (44) and that $\rho_{\text{end}} \simeq \bar{V}_{\text{end}}$. Therefore, using that at reheating $\rho \simeq \rho_r = \frac{\pi^2}{30} g_* T^4$, the reheating temperature is

$$T_{\text{reh}} \simeq \frac{1}{\sqrt{\pi}} \left(\frac{30}{g_*}\right)^{1/4} (\Omega_r^{\text{end}})^{3/4} \, \bar{V}_{\text{end}}^{1/4} , \tag{46}$$

where $g_*$ is the number of effective relativistic degrees of freedom at reheating. Putting in the numbers, we find $T_{\text{reh}} \sim 1\,\text{TeV}$. However, Figures 2 and 3 suggest that, immediately after inflation, the energy density of the field does not fall as rapidly as $a^{-6}$. This means that the radiation density takes over after the above estimate, corresponding to a somewhat lower reheating temperature.

The appropriate $\xi$ so that we can have successful coincidence is $\xi \sim 10^{-5}$. In order for the running in Equation (14) to result in this value we find that we need $\beta \approx -0.101$, which is rather reasonable. From Equation (40) we can estimate $\varphi_0/m_P \approx e^{-1/2\beta}|\mu| \simeq 840$. Then, Equation (11) suggests $\phi_0/m_P \simeq 80$, as can also be seen in Figure 5.

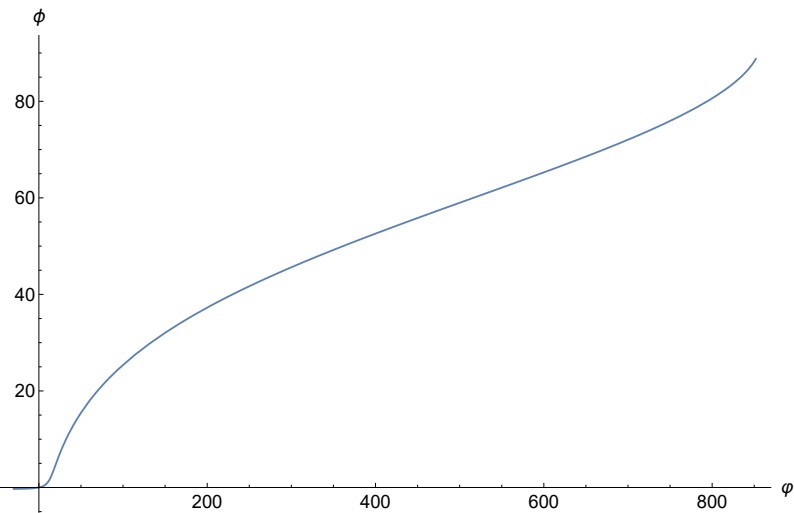

**Figure 5.** The relation of the canonical $\phi$ with the non-minimal $\varphi$ when $\kappa \simeq 0.3$ and $\xi \sim 10^{-5}$.

With these values we see that $\sqrt{\xi}\,\phi_0/m_P \simeq 0.25 < 1$. According to the discussion after Equation (13), the potential approximates a decaying exponential of strength $\kappa$. Since $\kappa < \sqrt{2}$, quintessence will approach the dominant attractor solution, for which the barotropic parameter is $w_\phi = -1 + \kappa^2/3$ [2]. With $\kappa = 0.3$ we obtain $w_\phi = -0.97$.

However, the approximation is not very good because $\sqrt{\xi}\,\phi_0/m_P$ is not very small. Indeed, using the parameter values in Table 1, for the dark energy barotropic parameter today we find

$$w_\phi^0 = w_{\text{DE}}^0 = -0.956 \quad \text{and} \quad w_a = -0.1596, \tag{47}$$

which satisfy the requirements in Equation (36) and will be observable in the near future. The above is an existence proof that our model works. We will attempt an exploration of the parameter space (which is a subset of the one shown in Figure 4) in Ref. [28]. The evolution of the barotropic parameters of the scalar field and the Universe after inflation is shown in Figure 6.

We see that the barotropic parameter of the Universe after equality is not exactly zero. In fact, we find that it peaks to almost $w \simeq 0.04$ at $z \simeq 200$. However, it reduces substantially for smaller redshifts and is very close to zero near $z \simeq 3$–5, which is when galaxy formation occurs, as shown in Figure 7. It would be interesting to investigate characteristic observational signatures of our scenario with respect to the growth of structures, but this is beyond the scope of this paper.

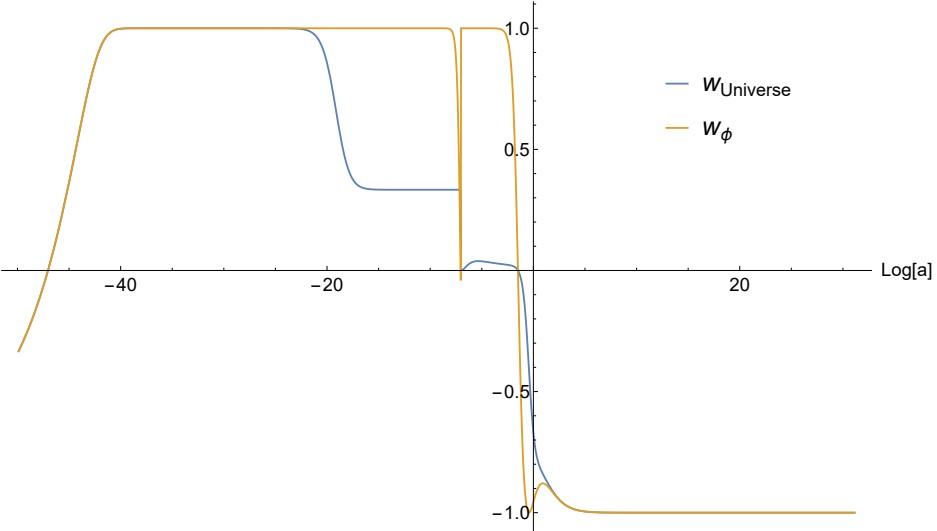

**Figure 6.** The evolution of the barotropic parameters of the scalar field (orange) and of the Universe (blue) after the end of inflation. We see that during kination both barotropic parameters are the same and equal to unity, because the scalar field dominates. After reheating, the barotropic parameter of the Universe reduces to 1/3, while the scalar field continues to be kinetically dominated with barotropic parameter $w_\phi = 1$. The field freezes briefly at radiation–matter equality, when its barotropic parameter is drastically reduced, while the Universe's barotropic parameter approximates zero. The backreaction due to matter does not allow the scalar field to stay frozen. Instead, it free-falls again with $w_\phi = 1$ until the present time, when it reduces drastically towards $-1$. The barotropic parameter of the Universe at present is found to be $-0.669$.

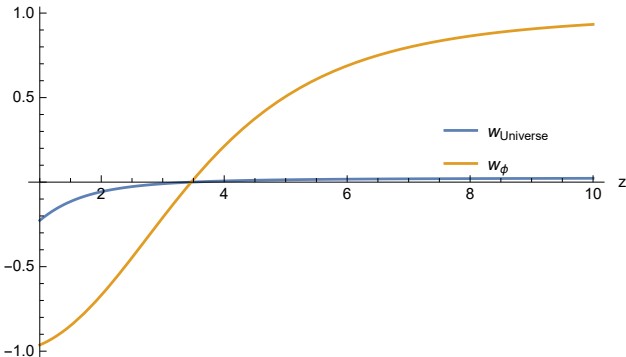

**Figure 7.** The evolution of the barotropic parameters of the scalar field (orange) and of the Universe (blue) as a function of redshift near the present. It is evident that $w \approx 0$ near $z \simeq 4$, when galaxy formation occurs.

## 8. Conclusions

We have investigated a model of quintessential inflation in the context of Palatini-modified gravity. We considered a non-minimally coupled scalar field and an $R^2$ contribution to the Lagrangian, both of which are rather modest modifications of gravity, frequently considered in the literature. The scalar potential of our non-minimal field is simply an exponential, which is well motivated in fundamental theory. The non-minimal coupling follows a mild logarithmic running, expected by renormalisation considerations, such that it is not the same during inflation and the present.

We find that our model can indeed successfully account for the observations of inflation and dark energy without any unphysical fine-tuning. The strength of the exponential potential is $\kappa \simeq 0.3$ and the non-minimal coupling runs from $\xi \sim 10^{-2}$ during inflation to $\xi \sim 10^{-5}$ during quintessence. The non-perturbative coupling of quadratic gravity is $\alpha \gtrsim 10^7$ (we consider $\alpha \sim 10^{11}$). The energy scale in our exponential potential turns out

to be $M \simeq 2 \times 10^{16}$ GeV, i.e., the scale of grand unification. The barotropic parameter of quintessence and its running are to be probed in the near future, e.g., by the EUCLID [37] and Nancy Grace Roman (former WFIRST) [38,39] satellites.

Our model leads to a long period of kination (with reheating temperature $T_{reh} \lesssim 1$ TeV). After kination the field freezes but soon it unfreezes again after equality (between matter and radiation), when the backreaction from coupling to matter kicks in. We find that the barotropic parameter of the matter era is affected in a diminishing way, such that it is approximately zero at the time of galaxy formation, as required. It is an open question whether its early values (almost 4% at redshift 200 or so) affect structure formation, in ways which could be an observational signature for our scenario.

**Author Contributions:** Conceptualization, K.D. and A.K.; methodology, K.D., A.K. and E.T.; software, S.S.L. and E.T.; validation, K.D., A.K., S.S.L and E.T.; formal analysis, K.D., A.K., S.S.L. and E.T. ; investigation, S.S.L. and E.T.; resources, K.D., A.K., S.S.L. and E.T.; data curation, S.S.L and E.T.; writing—original draft preparation, K.D.; writing—review and editing, K.D.; visualization, K.D. and S.S.L.; supervision, K.D.; project administration, K.D.; funding acquisition, K.D., A.K. and E.T. All authors have read and agreed to the published version of the manuscript.

**Funding:** K.D. was partly funded by the Science and Technology Research Council under STFC grant: ST/T001038/1. A.K. was funded by the Estonian Research Council grant PSG761. E.T. was funded by the Estonian Research Council grants PRG803, MOBTT5 and PRG1055. A.K. and E.T. were also funded by the European Regional Development Fund CoE program TK133, S.S.L. was supported by the Faculty of Science and Technology of Lancaster University.

**Data Availability Statement:** Data sharing not applicable. No new data were created or analyzed in this study. Data sharing is not applicable to this article.

**Conflicts of Interest:** The authors declare no conflicts of interest.

## Notes

[1]   See Ref. [24,25] for recent reviews on $F(R)$ gravity and Ref. [26] for a recent study of $F(\varphi, R)$ phenomenology.

[2]   This is because of the global scale invariance of the $R^2$ term, which is true in both the metric and the Palatini formalisms. We are thankful to the referee for pointing this out.

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
