# Peer review of "Modelling Quintessential Inflation in Palatini-Modified Gravity"

_galaxies, doi:10.3390/galaxies10020057_

Round 1

Reviewer 1 Report

The authors consider initially an F(R,\phi) theory in the Jordan frame and after transforming it to the Einstein frame, they study the resulting theory in the Palatini formalism. I have several comments prior to publishing the article:

  1. When transforming in the Einstein frame, one would expect a second dynamical degree of freedom generated from the R^2 term. Instead I can see only one scalar, why is that so?
  2. The authors indicate that post-inflationary a kination era occurs, thus the total EoS parameter of the total effective fluid corresponds to that of a stiff fluid with w=1. Hence, this could directly affect the inflationary e-foldings number and the total duration of the inflationary era, see Ref [1] in the authors reference list. Thus it would be interesting to see how the inflationary phenomenology is affected by this initially stiff reheating era.
  3. Some fundamental reviews on f(R) gravity are missing, for example the recent 1705.11098, and 1011.0544.  These should be cited appropriately.  Also recently f(R,\phi) models phenomenology of inflation in the Jordan frame was performed in 2108.04050, which is similar in spirit with the Lagrangian the authors use, namely Eqs. 1 and 2
  4. Some motivation for using the Lagrangian 1 should be given, for example the R2 term is one of many quantum correction terms to conformally coupled scalar field inflation, or minimally coupled scalar field, see 1507.06308. I think the authors should discuss this perspective for strengthening their motivation for the study.

After these comments are appropriately addressed, the article can be accepted for publication.  

Author Response

Please see attachement.

Reviewer 2 Report

Report on "Modelling Quintessential Inflation in Palatini Modifed Gravity", by K. Dimopoulos et al.  

The objective is to describe the inflational scenerario at the start of the universe (cosmic inflation), how the inflation is cut-off, and reemerges in a later epoch of the universe, observed as the current acceleration of the universe (or quintessential inflation). The formulation used is based on Palatini. The Lagrangian has R**2 contributions and includes a coupling to a scalar field, a standard approach. The equations of motion are derived and solved. Finally, the dynamics of the inflation and the dynamics of the expansion of the universe are obtained. The theory presented describes a possiblesceneraio of unification of the early cosmic inflation and the current acceleration of the universe. Possible connections to observation are roughly discussed

The procedure is solid and gives an interesting overview on the current work done so far.

The manuscript is well written and gives detailed explanations, which enables to follow the steps. Furthermore, the results presented are of interest to the community.

I recommend this manuscript for publication in its present form.

Reviewer 3 Report

This is a good paper on the subject of scalar-driven cosmic acceleration which explain primordial inflation and also the current phase of cosmic acceleration. The authors employ a model, defined by equations (1-2) and (14), in which there is a non-minimally coupled scalar with R + R^2 gravity in the Palatini formalism. The authors provide a very clear analysis of this model, and adjust its parameters to achieve the correct scalar amplitude and scalar spectral index, with an acceptably low tensor-to-scalar ratio and an acceptable dark energy equation of state.

I don't much care for the authors' model, but that is a matter of taste. I can honestly say that I learned some things from the authors' analysis, and that is sadly not often true of papers in this field. The only major improvement I recommend is that the authors provide error-bars for the quantities in equation (47), based on the full range of parameters shown in the yellow band of Figure 4. Otherwise it is tough for the reader to judge how well up-coming measurements of the two parameters can falsify the model.

Two minor issues are:

(1) On page 6, the 2nd line after equation (19), the authors should change ``dominant of not'' to ``dominant or not''.

(2) T the end of that same paragraph, the authors might want to comment that the global scale invariance of the R^2 term is what causes the parameter \alpha to drop out of the trace. This is true even in the Palatini formalism.
